



# Mineralization of autochthonous particulate organic carbon is a fast channel of organic matter turnover in Germany's largest drinking water reservoir

Marlene Dordoni [1], Michael Seewald [2], Karsten Rinke [2], Robert van Geldern [1], Jakob Schmidmeier [1], Johannes A.C. Barth [1]

[1] Friedrich- Alexander-Universität Erlangen-Nürnberg (FAU), Department of Geography and Geosciences, GeoZentrum Nordbayern, Schlossgarten 5, Erlangen, 91054, Germany
[2] Helmholtz Centre for Environmental Research-UFZ, Brueckstrasse 3a, D-39114, Magdeburg, Germany

*Correspondence to:* Marlene Dordoni (marlene.dordoni@fau.de)

**Abstract.** Turnover of organic matter (OM) is an essential ecological function in inland water bodies and relevant for water quality especially for the potential of dissolved organic carbon (DOC) removal as well as due to emissions of $CO_2$. We investigated various sources of OM including DOC, autochthonous particulate organic carbon (POC), allochthonous particulate organic carbon (ExtPOC), and sedimentary matter (SED) in a temperate drinking water reservoir (Rappbode Reservoir, Germany) with respect to carbon isotope ratios and concentration dynamics. For this purpose, we focused on the metalimnion and the hypolimnion, where respiratory turnover is expected to be dominant and hardly disturbed by atmospheric exchange. The observation period of nine months covered a complete stratification period of the water body. Dissolved inorganic carbon (DIC) concentrations and its isotopes ($\delta^{13}C_{DIC}$) were considered together with isotope data of DOC and POC ($\delta^{13}C_{DOC}$ and $\delta^{13}C_{POC}$) as input parameters for mass balances. DIC concentrations ranged between 0.30 and 0.53 mmol $L^{-1}$, while $\delta^{13}C_{DIC}$ values were between -15.1 and -7.2 ‰ versus the VPDB (Vienna PeeDee Belemnite) standard. Values of $\delta^{13}C_{DOC}$ and $\delta^{13}C_{POC}$ ranged between -28.8 and -27.6 ‰ and between -35.2 and -26.8 ‰, respectively. Isotope compositions of sedimentary material and allochthonous POC were inferred from the literature with average values of $\delta^{13}C_{SED} = -30.7$ ‰, and $\delta^{13}C_{ExtPOC} = -31.8$ ‰. Comparison of DIC concentration gains and stable isotope mass balances showed that autochthonous POC from primary producers was the main contributor to DIC increases, while contributions from DOC, ExtPOC and SED played a minor role. OM turnover rates, i.e. the conversion of organic carbon towards DIC, calculated with our isotope approach were within the range for oligotrophic water bodies (0.01 to 1.3 µmol $L^{-1}$ $d^{-1}$). Some higher values in the metalimnion are likely due the availability of settling POC from the photic zone. Samples from a Metalimnetic Oxygen Minimum (MOM) showed a clear dominance of respiration over photosynthesis through bacterial degradation of autochthonous POC. These high turnover rates further highlight a close link with planktonic biological assemblages. Our work shows that respiration in temperate lentic water bodies largely depends on autochthonous POC production as a major carbon source.

## 1. Introduction

Among the carbon phases in terrestrial surface waters, dissolved inorganic carbon (DIC) is usually most abundant, with concentrations that range from 0.1 mmol $L^{-1}$ to more than 1.0 mmol $L^{-1}$ (Cole and Praire, 2014). This wide range



of concentrations is closely linked to metabolic processes, because DIC is at the same time a reagent for photosynthesis and a product of respiration. Respiration rates vary with temperature, organic matter load and nutrient availability and phytoplankton assemblages (Gattuso et al., 2002; Hanson et al., 2003; Pace and Prairie, 2005; Wu and Chen, 2011; Mazuecos et al., 2015). A fundamental contribution to DIC budgets in water systems is represented by atmospheric and soil $CO_2$. Because of its high solubility in waters, this gas plays a major role in the shallower parts of lentic aquatic

ecosystems from which it either degasses (in case of heterotrophy) or it gets dissolved from the atmosphere (autotrophy) into the surface layer (Wetzel, 1984; Cole et al., 2007; Tranvik et al., 2009; Raymond et al., 2013; Koschorreck et al., 2017; DelSontro et al., 2018).

Other carbon phases in waters are represented by organic matter (OM). Organic carbon compounds result from decomposition processes of dead organic matter within (i.e. autochthonous) or outside (i.e. allochthonous) a water

body (Kritzberg et al., 2004). Almost all authochnous material in the pelagic zone of a water body consists of dissolved organic carbon (DOC) and dead particulate organic carbon (POC) and, to a smaller extent, dissolved organic carbon generated by leaching (Wetzel, 1984). Only a small fraction accounts for living biota (Kawasaki and Benner, 2006). Allochthonous OM is primarily of terrestrial plant origin and is transported to lentic systems by runoff as dissolved or particulate phase (ExtDOC and ExtPOC). ExtDOC and ExtPOC phases can also derive from atmospheric inputs such

as transport of dust by wind (Willey et al., 2000). In most cases, ExtDOC and ExtPOC concentrations are subject to seasonal variations. Residual autochthonous and allochthonous carbon that reach the bottom of a lentic water body can accumulate and generate sedimentary organic carbon (SED). This carbon burial in sediments is one of the main mechanisms of carbon sequestration within water bodies (Regnier et al., 2013).

DOC and POC in lentic waters are of primary importance in terms of water quality. DOC absorbs light and may inhibit

photosynthesis (Karlsson et al., 2009). This affects dissolved oxygen (DO) and DIC dynamics (Blough, 2001; Schindler, 2004). In many cases, DOC can also cause browning of waters that associates with numerous economic disadvantages. In waters used for drinking water supply, elevated DOC contents do not only pose aesthetic concerns. For instance, DOC reactions with chlorine during disinfection can produce numerous harmful by-products during water treatment (Karst et al., 2004; Bond et al., 2014; Fisher et al., 2017). Therefore, the evaluation of OM turnover,

especially with respect to DOC, is of critical importance for water quality management of surface waters used for drinking water production. On the other hand, the rate of POC turnover may also affect sedimentation rates and therefore control the amounts of carbon deposited in sediments (Azam et al., 1983; Keaveney et al., 2020). This may lead to excessive losses of methane from lentic water bodies, which is relevant for greenhouse gas production (Bastviken et al., 2010; Huang et al., 2019; Jansen et al., 2022).

Also the mineralisation of OM into DIC has implications on greenhouse gas dynamics as it releases $CO_2$ into the water, thus causing increases in DIC (Sun et al., 2016). In lakes that are poorly affected by inorganic carbon sequestration processes such as photosynthesis or calcite precipitation (Khan et al., 2022), increases of DIC concentrations may lead to $CO_2$ degassing from the water column. From this perspective, lentic water bodies are not only reactors for carbon turnover but also sources of $CO_2$ (Åberg et al., 2004; Cole et al, 2007). This highlights the

need for studies that investigate OM contributions to the DIC pool in aqueous systems.



Carbon turnover in aqueous systems can be investigated by means of concentrations and carbon stable isotopes ($\delta^{13}C$) of DIC, DOC, and POC (MacKenzie et al., 2004; Schulte et al., 2011; van Geldern et al., 2015). A common scheme is that respiration causes simultaneous decreases of $\delta^{13}CDIC$ with increases of DIC concentration in water (Stiller and Nissenbaum, 1999; Gammons et al., 2014). This is related to the fact that OM is usually $^{12}C$-enriched and therefore

has more negative $\delta^{13}C$ values when compared to its inorganic counterparts. During OM degradation, this more negative isotope signal is transferred to the DIC pool. Therefore, mass balances that combine carbon concentration data with isotopes may serve as useful tools for the study of carbon turnover in aqueous systems (Barth et al., 2017).

We investigated the turnover of OM into DIC with data from the temperate Rappbode Reservoir in Germany. With this, we aimed to constrain preferential turnover of one or the various sources of POC (allochtonous, autochtonous,

sedimentary) or DOC. A further objective was to define turnover rates and to outline in which compartments of the reservoir they might be highest. We also applied this approach to a Metalimnetic Oxygen Minimum (MOM) in the reservoir. Here, carbon turnover may be more pronounced as it is known for its strong depletion in dissolved oxygen (Shapiro, 1960; Nix, 1981; Kreling et al., 2017; Dordoni et al., 2022). The Rappbode Reservoir was ideal for this study because in recent years its logistic settings helped to develop it into an increasingly acknowledged large-scale

laboratory of ecological behaviour for man-made lentic water bodies (Kong et al., 2019; Wentzky et al., 2019; Herzsprung et al., 2020; Mi et al., 2020).

## 2. Methods

The Rappbode Reservoir in the Harz Mountains is the largest drinking water reservoir in Germany, with a surface area of 3.9 km$^2$ and a maximum volume of 0.113 km$^3$. The reservoir also has considerable importance for hydropower

production (Rinke et al., 2013). It receives water from two streams, the Hassel and the Rappbode. Two pre-reservoirs with a total volume < 0.002 km$^3$ are set on the path of these two rivers and are described in detail by Friese et al. (2014). Another input to the Rappbode Reservoir is an artificial water transfer gallery from the Königshütte Reservoir.

Sampling campaigns at the Rappbode Reservoir were carried out between March 2020 and December 2020 at a central location of the water body (51° 44´ 19” N, 10° 53´ 30” E, 420 m a.s.l.; Fig. 1). This monitoring station is located at a

distance of around 515 meters from the main dam. Sampling campaigns took place at least every two weeks. For this work, we investigated depths at 13, 16, 22, 40 and 65 meters below the water surface. Samples were collected with a LIMNOS-Watersampler$^{TM}$ (HYDRO-BIOS) in 1 L airtight amber-glass borosilicate bottles (DURAN$^{TM}$). They were prepared for laboratory analyses within one hour after sampling.

In the field, measurements of temperature and pH were performed on each depth-specific sample by a digital multi-

parameter instrument (HQ40d; Hach Company, Loveland, CO, U.S.A. ). Water samples for dissolved inorganic carbon (DIC) and dissolved organic carbon (DOC) were filtered through 0.45 µm pore-size syringe disk filters (Minisart HighFlow PES, Sartorius AG, Germany) into 40-mL amber-glass vials closed airtight by butyl rubber septa. These vessels conform to US Envrionmental Protection Agency (EPA) standards and were pre-poisoned with 20 µL of a supersaturated mercuric chloride (HgCl$_2$) solution to avoid secondary microbial activities after sampling.

Duplicate samples were stored in the dark at 4 °C.





In the laboratory, water samples were analyzed for carbon stable isotopes of DIC ($\delta^{13}C_{DIC}$) and DOC ($\delta^{13}C_{DOC}$) by an Aurora 1030W TIC-TOC analyzer (OI Analytical, College Station, Texas) that was coupled in continuous flow mode to a Thermo Scientific Delta V plus isotope ratio mass spectrometer (IRMS). Concentrations were determined from the signal of the OI internal non-dispersive infrared sensor (NDIR) and a set of calibration standards. All stable isotope

values in our work are reported as δ-notations versus the standard reference for carbon isotopes (Vienna Pee Dee Belemnite – VPDB) according to:

$$\delta = (R_{sample}/ R_{reference}) -1 \qquad (1)$$

where R is the molar ratio of the heavy and light carbon isotopes (i.e. $^{13}C/^{12}C$). These results were multiplied by 1000 to express them in per mille (‰). Standard deviations of both $\delta^{13}C_{DIC}$ and $\delta^{13}C_{DOC}$ measurements were better than

± 0.3 ‰ (1-σ).

Particulate organic carbon (POC) from water samples was collected on pre-heated (400°C) glass fibre filters (GFF) with a pore size of 0.4 µm (Macherey Nagel GF-5). The GFF were freeze-dried and subsequently pulverized using a ball mill (Retsch CryoMill). The resulting powder was fumigated by concentrated HCl in a desiccator for 24 hours to eliminate possible carbonate particles on the filters. Afterwards, the sample was stored for 1 hour at 50 °C to allow

degassing remaining acid fumes. Grinded and de-calcified GFF sample material was then weighed into tin capsules for isotope analyses. Samples were analyzed for their $\delta^{13}C_{POC}$ signals using a Costech Elemental Analyzer (ECS 4010; Costech International, Pioltello, Italy; now NC Technologies, Bussero, Italy) in continuous flow mode coupled to a Thermo Scientific Delta V plus isotope ratio mass spectrometer (ThermoFisher Scientific, Bremen, Germany). The standard deviation of these measurements was was better than ± 0.3 ‰ (1-σ).

The vertical division of the water column into the compartments (epilimnion, metalimnion and hypolimnion) was arranged according to temperature data (supplementary material, Fig. S1).

The mass balance for carbon turnover calculations only works for environments where respiration is the dominant process. This is because in the epilimnion the $\delta^{13}C_{DIC}$ signal is influenced by photosynthesis, and also by degassing of $CO_2$ to the atmosphere. Both these processes shift the $\delta^{13}C_{DIC}$ values towards more positive values in patterns that are

difficult to predict and depend on the type of algae community and magnitudes of $CO_2$ degassing. These uncertainties render a closed mass balance approach difficult for the epilimnion (Barth et al., 1998; van Geldern et al., 2015). Therefore, we restricted mass balance calculations for OM turnover to the metalimnion and the hypolimnion. The isotope mass balance equation relies on the determination of molar contibutions from OM to DIC pool as follows:

$$n_{fromOM} = n_t \times \frac{(\delta^{13}C_t - \delta^{13}C_s)}{(\delta^{13}C_s - \delta^{13}C_{OM})} \qquad (2)$$

where

$n_{fromOM}$ is the molar C-contribution of OM to the DIC pool

$n_t$ is the molar concentration of the DIC at the time when the water column was homogenised by the lake turnover at the beginning of the year (time 0, i.e. 17 March 2020)





and

$\delta^{13}C_t$ is its correspondant carbon isotope composition

$\delta^{13}C_s$ is the isotope composition of DIC of any sampling event later than time 0

$\delta^{13}C_{OM}$ is the isotope composition of the considered OM source material that was turned into DIC.

The organic matter (OM) sources that we studied were DOC, POC, SED and particulate organic carbon of allochthonous origin (ExtPOC). Their isotope composition ($\delta^{13}C_{OM}$) was the variable that was replaced for each

sample calculation in equation (2). We used field data for all the variables of the equation, except for $\delta^{13}C_{SED}$ and $\delta^{13}C_{ExtPOC}$. For the latter two, we used data from the literature that corresponded to -30.7 ‰ and -31.8 ‰, respectively (Barth et al., 2017). One requirement for using this method is the increase in DIC from time 0 to each subsequent sampling event. We tested the validity of equation (2) via an error propagation approach (Ku, 1966; Kretz, 1985):

$$S = \sqrt{\left(\frac{\partial n_{fromOM}}{\partial n_t}\right)^2 \cdot sn_t{}^2 + \left(\frac{\partial n_{fromOM}}{\partial \delta^{13}C_t}\right)^2 \cdot s\delta^{13}C_t{}^2 + \left(\frac{\partial n_{fromOM}}{\partial \delta^{13}C_s}\right)^2 \cdot s\delta^{13}C_s{}^2 + \left(\frac{\partial n_{fromOM}}{\partial \delta^{13}C_{OM}}\right)^2 \cdot s\delta^{13}C_{OM}{}^2} \quad (3)$$

Where S is the total standard variation and s refers to the standard variation of each sampling date. More information about this method is available in supplementary material.

The evaluation of OM turnover into the DIC pool is limited to the lower metalimnion and the hypolimnion of the Rappbode Reservoir because only in these zones we obtained a good correlation between DIC and its isotopes (supplementary material, Fig. S2). For comparison to the isotope mass balance (Eq. 2) we also determined DIC gains

with concentration differences from time 0 ($n_{s-t}$). Here we aimed to study which correlations would be closest to a 1:1 line in order to narrow down the most plausibel source of OM turnover.

We also calculated OM to DIC turnover rates in the metalimnion and in the hypolimnion. For the metalimnion, we additionally separated the layer showing the minimal oxygen concentration (Metalimnetic Oxygen Minimum, MOM) and calculated MOM-specific turnover rates. Our results are expressed in µmol L⁻¹ d⁻¹ for individual sampling dates

with their time differences between time 0 (17 March 2020) up to a maximum of 259 days (i.e. 08 December 2020). In order to evaluate OM seasonal turnover rates, we subdivided our database according to Wang et al. (2021). Spring ranged from 17 February to 11 June, summer from 12 June to 14 September, autumn from 15 September to 5 December and winter from 6 December to 16 February.

### 3. Results

Figure 2 shows DIC profiles within the studied time period divided into seasons. DIC concentrations generally increased from 17 March 2020 (time 0) over the course of the study period. The spring profiles showed the smallest range of DIC concentrations (0.29 mmol L⁻¹ to 0.36 mmol L⁻¹).

Concentration differences between depth profiles were smallest in summer. However, they showed a pronouced increase in DIC concentration with respect to time 0, with values ranging from 0.33 to 0.58 mmol L⁻¹ (Fig. 2). A



sample from 22 meters depth on 04/08/2020 had an exceptional high DIC content, whereas one DIC decrease was recorded at 16 meters depth on the 18/04/2020.

Samples from autumn had the highest DIC concentrations with values between 0.38 and 0.59 mmol L$^{-1}$ (Fig. 2). A noteworthy feature for autumn and winter was the presence of very high DIC concentrations in samples from 13 and 16 metres depth on 29/09/2020. They showed values of 1.00 and 0.94 mmol L$^{-1}$, respectively (Fig. 2).

For comparison to the isotope mass balance (Eq. 2) we also determined DIC gains with concentration differences from time 0 (n$_{s-t}$). These results were plotted against the DIC concentration gains as calculated by equation (2) (Fig. 3). With this analysis, in situ produced POC showed the best linear regression that was closest to a 1:1 line (Tab. 1). This was even clearer when only the hypolimnion was considered, with an angular coefficient of the regression line for POC being 1.00. On the other hand, in the metalimnion the angular coefficient for DOC was reasonably close to the

one of POC with values of 1.15 and 1.12, respectively. The MOM was considered separately and also showed a clear predominance of POC input as the OM sources with an angular coefficient of 1.05.

DIC production rates from POC turnover as the dominant OM input in the water column of the Rappbode Reservoir are reported in figure 4. They ranged from 0.1 to 1.3 µmol L$^{-1}$ d$^{-1}$. The metalimnion and the upper part of the hypolimnion showed higher rates, up to 1.3 µmol L$^{-1}$ d$^{-1}$. The metalimnion showed less variations than the

hypolimnion and ranged from 0.3 to 1.1 µmol L$^{-1}$ d$^{-1}$. The highest DIC production rates were found among the MOM samples, which complies with the fact that oxygen depletion is most intense in this layer. DIC rates for the hypolimnion ranged from 0.1 to 1.3 µmol L$^{-1}$ d$^{-1}$, with the highest variance at a water depth of 22 meters (0.2 to 1.3 µmol L$^{-1}$ d$^{-1}$). Samples from 40 m depth had the lowest DIC productivity with values below 0.2 µmol L$^{-1}$ d$^{-1}$. With one exception at 40 m depth, data from 65 meters depth had higher rates but covered a narrower range of variation (0.5 to 0.7 µmol L$^{-1}$ d$^{-1}$).

Data from springtime had the highest spread of POC turnover rates for each depth except for 13 meters. In this depth, MOM waters exceeded the springtime range of variation. Also, samples from summer and autumn showed higher turnover rates than those of spring. Overall, the highest carbon turnorver rates to DIC were found in springtime and during summer, when the MOM developed. The complete database is available in the supplementary material.

**4. Discussion**

At the Rappbode Reservoir, DIC concentrations througout the water column were almost homogeneous during the mixing period. This is in accordance with the theory of biogeochemical uniformity during mixing periods of lentic water bodies (Wetzel, 1984). DIC values at this starting point were lower when compared to subsequently collected samples because the water column was closest to equilibration with the atmosphere and, due to lower temperatures,

metabolic processes were still low. The following continuous increase of DIC below the photic zone in the metalimnion and the hypolimnion are most plausibly due to respiration and to reduced exchanges between the compartments due to stratification. This can best be shown in the hypolimnnion, where the counteracting effects of photosynthesis and CO$_2$ degassing (processes that both reduce DIC contents) are minimal.





DIC concentration increases near the bottom of the lake (65 m) are usually attributed to contributions from processes involving organic turnover within the sediments (Wetzel, 1984). On the other hand, higher DIC concentrations found in the metalimnion are related to respiration processes that link to OM input from above. Notably, the MOM samples had the highest DIC concentrations, with values up to 1 mmol L$^{-1}$. This matches the findings of Giling et al. (2017) who also found that the metalimnion may act as a deicisive metabolic hospot in oligotrophic water bodies. Additionally, MOM samples also had the highest POC and DOC concentrations (cf supplementary material), which

commonly mark zones close to phytoplankton assemblages (Wetzel et al., 1972). This is a typical scenario after the collapse of an algae bloom, when settling detritus becomes directly subject to decomposition and accelerates respiratory turnover. This hypothesis is supported by our data that show predominant autochthonous POC consumption. Temperature-related viscosity changes typical of the metalimnion may also have enhanced the residence time of organic matter. Altogether, these findings suggest a relation between the MOM and a productive biotic

assemblage in vertical proximity above the MOM. This connection is supported by data from the literature (Kreling et al., 2017; McDonald et al., 2022) and was already highlighted independently by previous studies with dissolved oxygen isotope insights on the Rappbode Reservoir (Dordoni et al., 2022).

It was not possible to evaluate OM contributions to the DIC pool for the epilimnion because no reasonable correlation between DIC concentration and its isotopes was found in this compartment (supplementary material, Fig. S2).

Although previous studies reported that respiration tends to exceed photosynthesis in the epilimnion of oligothrophic water bodies (del Giorgio and Peters, 1993; del Giorgio et al., 1997; Duarte and Agustí, 1998), this result was expected due to the intense influence of photosynthesis and $CO_2$ degassing to the atmosphere on $\delta^{13}C_{DIC}$. On the one hand, photosynthesis generates more positive $\delta^{13}C_{DIC}$ values (Ahad et al., 2008; Wachniew, 2006; Gammons et al., 2011). On the other, shifts of $\delta^{13}C_{DIC}$ towards more positive values also result from $CO_2$ degassing to the atmosphere.

Therefore, OM turnover calculations could not be corrected for these counteracting and less systematic effects. Nonetheless, it is likely that OM turnover rates in the epiliminion are higher than in the other two compartments because the photic zone directly offers fresh authochtonous POC material for turnover.

Note that regressions for the metalimnion also leaves room for uncertainty (Table 1). This is because the photic zone at the Rappbode Reservoir can extend down to metalimnetic depths, where residual influences of photosynthesis may

complicate the relationships between OM and DIC as described for the epilimnion.

The isotope approach used to assess the turonver of OM within the metalimnion and the hypolimnion of the Rappbode Reservoir confirmed that POC is the main OM contributor to the DIC pool. This was shown by the closest fit to the 1:1 line of the correlation between DIC gains as calculated from differences between DIC concentration from 17 March 2020 (time 0) and each sampling event and the results from the mass balance in equation (2). This good fit

serves as a good indication that autochthonous POC is indeed the most favorable OM used for respiratory turnover into DIC.

Calculation-related uncertainties may have been higher when we used average $\delta^{13}C$ input values of SED and ExtPOC from the literature without any variance to constrain their contribution to the DIC pool. However, in the absence of field data, our approximation should be sufficient to provide an accurate estimation. The reason why allochtonous-



related OM does not seem to contribute significantly to the DIC pool in the reservoir might be that its mass is insufficient to exert enough impact on the whole carbon budget. This suggests either low inputs from the tributaries and the catchment or low turnover of this carbon phase within the reservoir. The latter option is more plausible as POC concentrations were often higher than expected by autochtonous POC concentrations and reached values of up to 1.05 mg L$^{-1}$ (supplementary material).

Despite its high concentrations in the water column, the influence of DOC seemed to have secondary influences on DIC increases. This was obvious by the regression lines in Fig.4. Although studies on rivers have proven that DOC can fuel metabolic processes in peat-dominated heterotrophic ecosystems (Thurman, 1985; Billett et al., 2010), the DOC pool in lentic oligotrophic water bodies consists primarily of carbon compounds that are older and more resistant to bacterial decomposition (Wetzel, 1984). As a result, bacteria prefentially consume autochthonous POC that consists 250 mainly of fresh material (Cole et al., 1984; Barth et al., 2017). Note, however, that if part of the DOC pool found in the Rappbode reservoir resulted from leaching of POC, it should have the same isotope composition as the original POC material itself. If this part of the DOC pool became turned over into DIC by respiration, we would not be able to differentiate from a direct POC input.

Overall, OM turnover rates into DIC were comparable to those found in other oligotrophic water bodies (Cole et al., 255 1984; Scavia and Laird, 1987; Carignan et al., 2000; Lammers et al., 2017). Therefore, our results seem to successfully approximate the expected setup of the aquatic environment and could be used as an alternative approach to quantify turnover rates of OM without the use of in situ incubation experiments.

Higher rates of POC turnover above 23 meters depth in figure 4 may have been caused by higher availabilities of POC produced by photosynthesis within the photic zone. This also indicates that OM turnover in this part of the water 260 column depends largely on freshly produced POC material that sinks downward and may decompose more rapidly due to high oxygen availability (Pace and Prairie, 2005). This is in agreement with the appearance of epilimnetic diatom blooms in early spring and phytoplankton blooms in the metalimnion of the Rappbode Reservoir from summer to early autumn (Wentzky et al., 2019). This explanation likely also applies to samples from a water depth of 40 metres, where the rate of respiratory carbon turnover depends primarily on the residence time of the detritus (Robarts, 265 1986). In support of this explanation, summer and autumn turnover rates are higher than the spring ones particularly at this depth (Fig. 4). Increased turnover rates for samples from 65 meters depth may appear as a contradiction. Such values likely result from detritus decomposition with carbon contributions from the sediments. Overall, the reason why spring turnover values are generally lower may lie in the combined effects of temperature, residence time, and nature of decomposing detritus. During springtime, most of the organic matter derived from diatom blooms is 270 sequestered within the epilimnion (Sommer et al., 1986) and the amount of detritus that can reach greater depths to become mineralized is low. Additionally, the heavy silica shells of diatoms decrease their residence time in the water column. Moreover, cold temperatures in spring reduce mineralisation rates.

Unlike the the rest of the metalimnion, MOM samples hardly seem to be influenced by photosynthesis and show a clear predominance of respiration. Also in this zone POC was determined as the main contributor to DIC pool, with 275 turnover rates that are the highest for summertime and among the highest in the whole database (Fig. 4). Once again,





this result suggests close relationships between metabolism of the auctochtonous phytoplankton community and its decay and the following emergence of the MOM (Shapiro, 1960). This finding also supports the interpretations by Kreling et al. (2017) and McDonald et al. (2022) of downward fluxes of POC promoting the development of MOM.

Overall, our study agrees with preliminary studies on the pre-reservoir dams in the Rappbode System (Barth et al., 2017). This suggests that similar OM turnover principles apply despite volume differences between the main water body and its pre-reservoirs.

## 5. Conclusions

Comparisons between DIC concentration differences and stable carbon isotope mass balances of OM turnover in an oligotrophic drinking water reservoir demonstrated dominant turnover of freshly produced POC. DOC turnover

seemed negligible, unless part of this pool was generated by leaching from the POC pool. This sort of DOC would be isotopically identical to its precursor POC material.

Calculated rates of POC turnover into DIC by respiration are typical of oligotrophic water bodies and could be assessed with our method without the use of in situ incubation experiments. Therefore, our approach may provide a promising alternative to complex incubation experiments. Nevertheless, such findings based on isotope mass balances should be

tested in parallel with data from incubations to evaluate and narrow down uncertainties.

Especially low turnover rates indicate the environmental fragility of the Rappbode Reservoir. This shows that if large amounts of carbon were added to the system, it likely could transfer only part of this load into DIC. Such a scenario could for instance occur with excessive algae blooms under different nutrient loads, washing in of carbon under different land use and climate conditions. In such a scenario, the current in situ respiration would likely be unable to

cope with the excessive C loads. This would lead to higher sedimentation and water browning. Likely modifications of metabolic balances would also compromise the water quality. Overall, a comparison between DIC production rates with DO depletion rates as described by Dordoni et al. (2022) should be evaluated in order to prove the consistency of both isotope approaches. Such investigations will be subject of future studies.

The above considerations are likely transferable to other temperate lentic systems, in which links between POC

turnover and associated DIC gains may operate at different rates. Overall, our results may help to improve water management strategies to help foster economic and environmental management of drinking water reservoirs.

## 6. Data availability

Data are available in the supplementary material.

## 7. Author contribution

MD performed the formal analysis and was responsible for investigation, data curation and visualization. MD and JB cured the conceptualization, methodology, validation, resources, and writing of the original draft. MS, KR and RvG



cured manuscript review and editing. KR and JB were responsible for project administration and funding acquisition. JB provided constant supervision.

## 8. Competing interests

The authors declare that they have no conflict of interest.

## 9. Acknowledgments

The authors would like to thank Karsten Rahn, Michael Herzog, Martin Wieprecht, Silke Meyer, Luisa Daxeder, and Lucas Heiß for assistance in the field during sampling, as well as Christian Hanke, and Anja Schuster for their commitment to laboratory analyses. We are grateful to the "Talsperrenbetrieb Sachsen-Anhalt" (TSB) for their support

and permissions. Funding: this work was supported by the Deutsche Forschungsgemeinschaft (DFG) [BA 2207/18-1 and RI 2040/4-1].

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



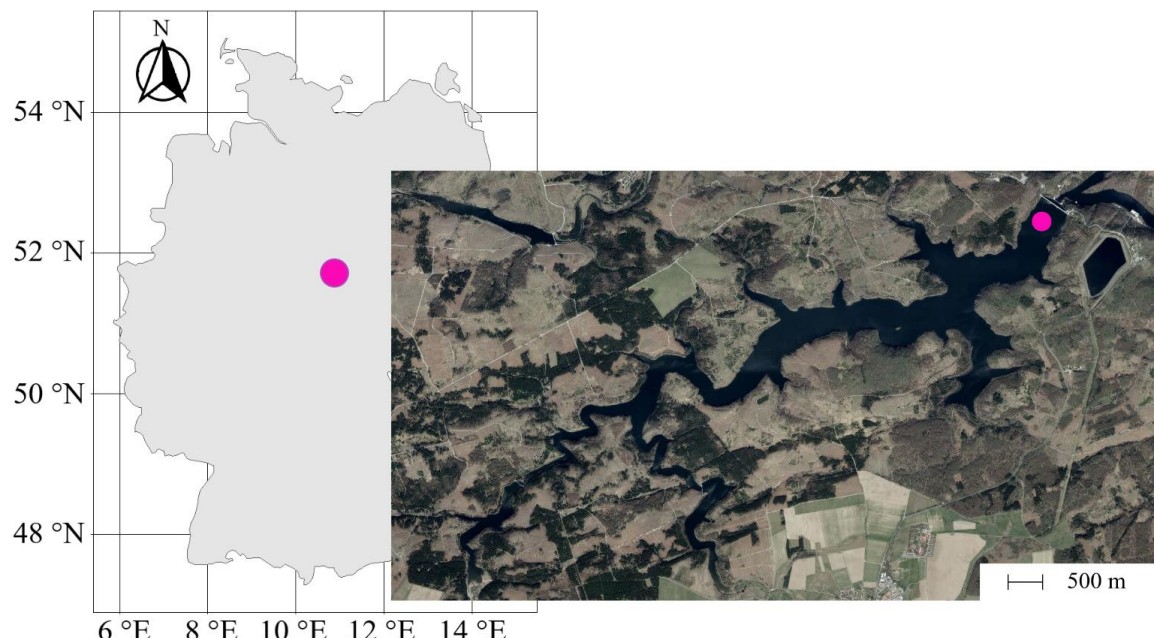

**Figure 1 : Position of the Rappbode Reservoir in Germany with the sampling location marked by a pink point in the main reservoir (source: ESRI).**

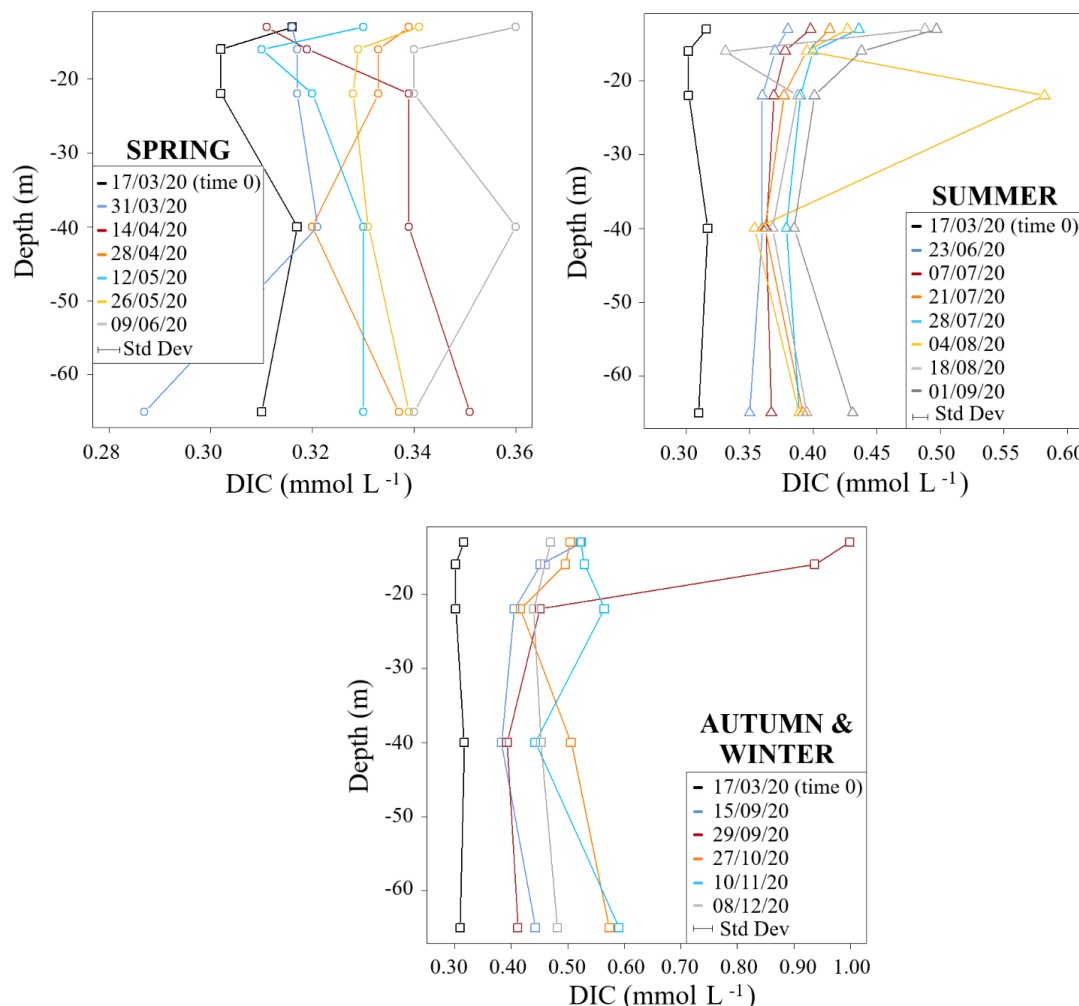


**Figure 2 : DIC seasonal concentration profiles. Standard deviations relative to each dataset (± 3%) are reported in the plot below the legend.**

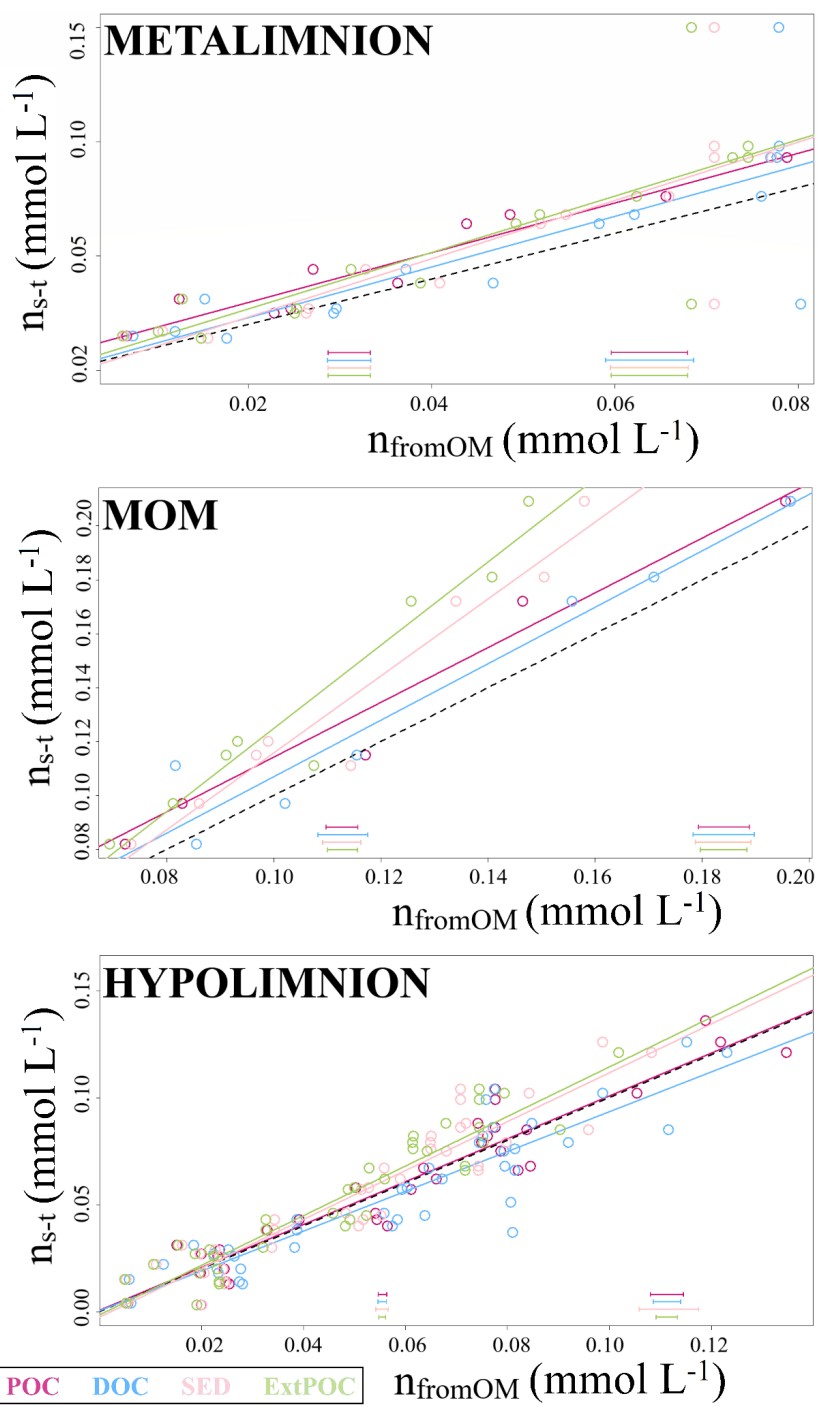

**Figure 3 : Molar gain by concentration from time 0 ($n_{s-t}$) and as calculated by mass balance with carbon stable isotopes ($n_{fromOM}$) in the metalimnion, MOM and hypolimnion of the Rappbode Reservoir. Datasets are reported in different colours according to the considered OM. Standard variations for each OM are reported within the plot.**





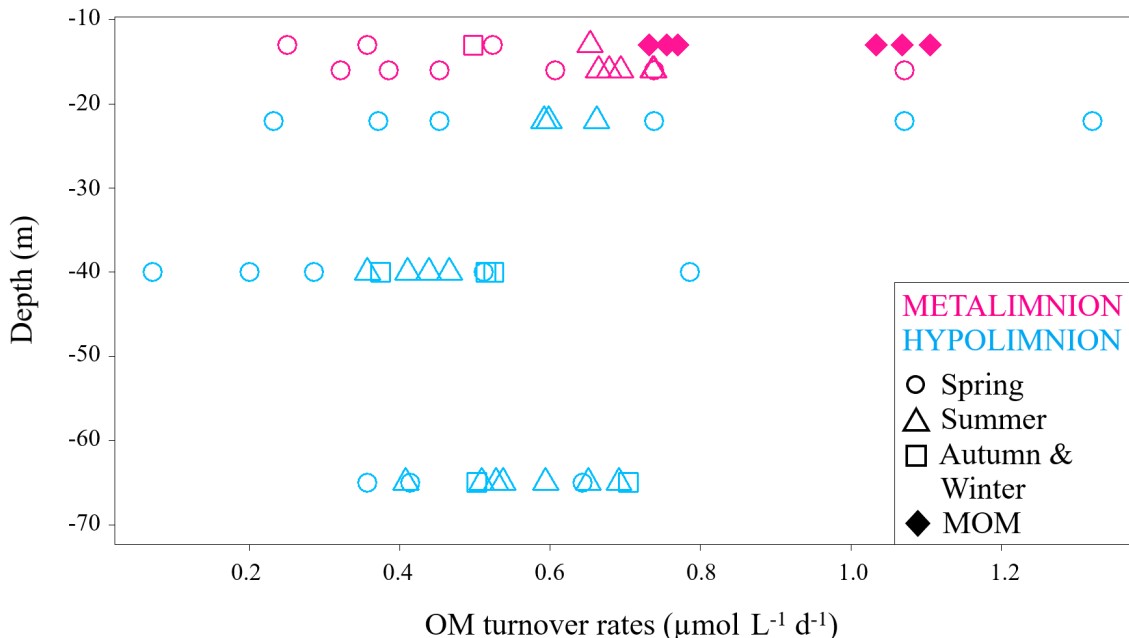

**Figure 4 : DIC production rates at specific depths of the hypolimnion and metalimnion during the studied time period, marked with different symbols according to sampling season. MOM samples belong to summertime and are displayed as diamond shapes for clarity.**



| OM input → | POC | DOC | SED | ExtPOC |
|---|---|---|---|---|
| *Metalimnion* | y = 1.12x + 0.01<br>$R^2$ = 0.95 | y = 1.15x + 0.00<br>$R^2$ = 0.85 | y = 1.35x - 0.00<br>$R^2$ = 0.88 | y = 1.34x + 0.00<br>$R^2$ = 0.88 |
| *MOM* | y = 1.01x + 0.01<br>$R^2$ = 0.95 | y = 1.09x - 0.02<br>$R^2$ = 0.94 | y = 1.43x - 0.03<br>$R^2$ = 0.94 | y = 1.54x - 0.03<br>$R^2$ = 0.94 |
| *Hypolimnion* | y = 1.00x + 0.00<br>$R^2$ = 0.91 | y = 0.93x + 0.00<br>$R^2$ = 0.82 | y = 1.41x - 0.00<br>$R^2$ = 0.88 | y = 1.16x - 0.00<br>$R^2$ = 0.89 |

**Table 1 : Equations of the regression lines for particulate organic carbon (POC), dissolved organic carbon (DOC), sedimentary material (SED), and allochthonous particulate organic carbon (ExtPOC) in the metalimnion, MOM and hypolimnion of the Rappbode Reservoir. The coefficient of determination ($R^2$) is reported below each equation. All p-values < 0.001.**
