# Peer review of "Mineralization of autochthonous particulate organic carbon is a fast channel of organic matter turnover in Germany's largest drinking water reservoir"

_Biogeosciences, 2022_

## Author Comment (AC1)

Erlangen, 24 August 2022

[Figure]

Dear Editorial Team of Biogeosciences, dear Reviewers,

thank you for the handling of the manuscript and for providing us your comments. We did our best to address the constructive criticisms.

Please find our answers in the table on the next pages. For the final submission, we will upload a new version of the manuscript with tracked changes and another one with all tracked changes removed for further processing.

With kind regards and on behalf of all co-authors

Marlene Dordoni (PhD student)

| Reviewer´s suggestion | Authors´ answer |
|---|---|
| **Reviewer #1** | |
| Overall, this is well-organized and clearly written manuscript. The data, discussion and conclusion are intuitive for the most part. That said, I do have some minor to moderate comments and suggestions that I think will improve the manuscript. I outline these below, with line numbers where appropriate. Once these changes have been made, I fully support publication of the manuscript in Biogeosciences. | We thank the reviewer. |
| L12-13: It may be better to use "phases" instead of "sources" here. Some abbreviation seem to make me confusing, e.g., POC and ExtPOC. Could the authors just use "auto-POC" and "allo-POC" to name these two differently-sourced POC. | Done. |
| L14-16: I am confused about which is the "this purpose". Does it refer to the first sentence of the Abstract? Could the authors change it to "For eliminating the influence of atmospheric exchange, we ……." ? Also, the Abstract could be improved to be more concise and logically clearer. | We modified the text accordingly. |
| L32: In the introduction part, a description of characterization of metalimnion and hypolimnion seems missing, because $CO_2$ exchanges from atmosphere and soil are also important sources of DIC. | We improved this part of the Introduction. |
| L110: The description of d13C could be simplified, as it is a basic parameter to the audience in this field. | Done. |
| L130: Instead, the isotope mass model could be explained in more details, e.g., how the equation (2) was deducedï¼ŸThe variations of d13CDOC and d13CPOC should be plotted in the main text, as they are important for the manuscript. | We deepened the details regarding equation (2) in the supplementary material to avoid an interruption of the flow of the main text. The graph with variations of $\delta^{13}C_{DOC}$ and $\delta^{13}C_{POC}$ was inserted into the main text. |
| L175: The Results part looks very discrete with 6 paragraphs, even some paragraphs are only composed of 3-4 sentences. Please revise the part to be more simplified and concise. | We merged Results and Discussion and divided the new section into more coherent paragraphs. |
| L231: Please add several sub-titles to discussion part to make it more readable. | The Discussion is now divided into subsections with headings. |
| | |

| Reviewer´s suggestion | Authors´ answer |
|---|---|
| **Reviewer #2** ||
| I think while this study is interesting the paper could do with restructure. The results are almost like bulletpoints and could be merged with the discussion to give it logic and context. | Results and Discussion are now merged into a single section with subsections that have individual headings. |
| I also am not convinced by the mass balance end-members. 13C isotopes vary so widely in freshwater that they overlap with terrestrial values. While the data show that there is differences between POC and DOC, there is enough overlap to reduce the conclusions that the authors state. I don't know if the endmembers the authors used for the calculations are right, and therefore, I'm not sure that the interpretation is correct. If the end-member is correct (the authors should jusitify this) and the description of the isotope methods was made clearer the paper could be accepted with minor revisions. If not, calculations should be revised and reconsidered after major revisions. | In our manuscript, we used endmembers that were already published (doi:10.1080/10256016.2017.1282478). At current, no further data more suitable for our study venue are available as input.

The endmembers for allochthonous POC and sedimentary POC are still in the range for C3 plants that are expected as starting material for organic matter input. The reason why the allochthonous POC and its related sedimentary POC range around – 31 ‰ (at the lower end spectrum of C3 plants) could be related to inputs from peaty material from a hardrock terrain that makes up the Rappbode Catchment. These relationships were explained at more detail in the Results and Discussion section. Carbon isotope compositions for autochthonous and allochthonous POC overlap only for some metalimnetic samples between March and May. This might be an additional reason why we were not able to constrain a definite OM source for DIC pool in the metalimnion. For the remaining data in the database, isotope compositions for autochthonous and allochthonous POC differed enough to differentiate these sources.
If the reviewer's concerns are directed at the data collected, we have reported the standard deviation of the relative instruments in our manuscript.
We agree with the reviewer that the POC and DOC correlations are similar for the metalimnion, and we further addressed this topic in the Results and Discussion. |
| I would prefer if the isotopes were plotted on their own with permil axis rather than the way they are plotted - it is hard to see the real scatter/overlap. | We agree that it is difficult to follow the manuscript with only the isotope results being used as input in figures 3 and 4 (now, figures 4 and 5). Also based on a request by Reviewer 1, in the new version we added contour plots of $\delta^{13}C_{DOC}$ and $\delta^{13}C_{POC}$ to the main text (new figure 3). |
| I also think that the terms that the authors use should be explained as they are not used in all countries. I have made comments in the attached pdf and hopefully these will help. | These comments were unfortunately unavailable to us and if possible we are happy to incorporate them at a later point. |
| Overall, I think the premise is good but I'm not convinced that the data shows what the authors conclude. some parts should be simplified and others should be explained properly. I think that the tables in the supp. information should actually be in the main article or isotopes plotted by their own. | We have modified the structure of the manuscript and given more space to the concepts reported. |

---

## Author Response (AR1)

Erlangen, 29 September 2022

[Figure]

Dear Sebastian Naeher and Editorial Team of Biogeosciences, dear Reviewers,

thank you again for the handling of the manuscript and for providing us with more precise comments. We did our best to address these constructive criticisms.

Please find our answers in the table on the next pages. The first round of comments to the reviewers was already uploaded on BG Discussion (https://bg.copernicus.org/preprints/bg-2022-154/#discussion). We also uploaded a new version of the manuscript with tracked changes and another one with all tracked changes removed for further processing.

With kind regards and on behalf of all co-authors

Marlene Dordoni (PhD student)

| Reviewer´s suggestion | Authors´ answer |
|---|---|
| **Associate Editor** | |
| In addition to the suggestions of the reviewers, I would like to see inclusion of more general information like maximum lake depth, lake mixing/overturn frequency and timing of mixing, productivity peaks, oxygen content (average in epi-/hypolimnion for different seasons?), potential methane ebullition from the sediments in the reservoir, etc. Such factors should be discussed, because they may impact (by how much?) the abundance and d13C composition of the organic matter in the lake and therefore influence the parameters that you investigated. | We have added a this information at the beginning of the Methods section including mixing turnover and productivity peaks as well as oxygen behaviour (Dordoni et al., 2022). The mentioned factors are important, however we point out that this study focuses only on one single point with detailed depth profiles at high frequency. We therefore suggest in the conclusions future studies that should inquire spatial and lateral heterogeneities. |
| Like reviewer 2, I am also not completely convinced about the determination of the end member composition. Especially using single values of d13C values of SED and extPOC from the literature (even if determined in the same reservoir) does not seem to be sufficient. Taken together with the limited discussion of above noted influencing factors, the high variability and potential large overlap of d13C in the reservoir and catchment is likely insufficiently captured. Therefore, more justification is needed to demonstrate the end members used are correct. | We searched the literature again and found three manuscripts that published data on $\delta^{13}C_{POC}$. One for sedimentary material (10.1007/s10533-022-00930-y) with a value of -31.1 ‰, and two on $\delta^{13}C_{POC}$ of allochtonous origin (10.1080/10256016.2017.1282478 and 10.1038/s41598-019-52288-1). Additionally, we inquired $\delta^{13}C_{POC}$ of allochtonous input in the studied catchment via personal communications. These $\delta^{13}C_{POC}$ data of river input were measured only sporadically and values were made available by the Helmholtz Centre for Environmental research (UFZ). They ranged between -28.7 ‰ and -32.5 ‰ and their average was -30.6 ‰. These data were added to the supplementary material. We used averages and most extreme values of these sources to show the variance of the inputs (new Fig.4 with green bands). This yields some overlap with results of $\delta^{13}C_{POC}$ of autochthonous material. However, when using average values we still obtain good separations. Note that this study did neither investigate sedimentary POC nor allochthonous POC at the same frequency we were sampling the reservoir with its depth profiles. Therefore, we have to rely on literature values and the ranges presented therein. |
| | |

| **Reviewer 2** | |
|---|---|
| When I said that the terms were not universal, I meant metalimnion/limnion and more details on the stratification terms. I wasn't clear, I meant that some of the stratification process and terms need to be explained as the process of the lake stratification for this lake is important and more details are needed. The authors speak about epi/meta/hypo but do not say anything about benthic/pelagic/littoral habitats, which also affects the isotopes, so this needs to be included too – horizonal spatial effects as well as vertical effects are also important for variations in 13C isotopes. I think the variability of the freshwater isotopes and the end-member they use from the lit. may not be correct – if they give more detail this may be enough to deal with this, but I'm not fully convinced. | As already answered to the Editor, we have added more information the manuscript that describes these points. We also referred to the supplementary material where we present a detailed plot of temperature distribution over time. This nicely shows one of the strongest controls of the reservoir turnover. This reservoir has a very small littoral zone and therefore its influence can be assumed as to be negligible. In terms of lateral heterogeneities, we have to admit that this work only investigated one profile at high temporal frequency. For this study we chose the best representative spot for studying the water column that was chosen also by other studies (10.1016/j.watres.2018.10.047, 10.1016/j.scitotenv.2022.156541, 10.1016/j.watres.2020.115701). In the conclusions, we recommend further testing of the lateral heterogeneities of the reservoir with isotope parameters. |
| | |

---

## Author Response (AR2)

Erlangen, 3 November 2022

[Figure]

Dear Sebastian Naeher and Editorial Team of Biogeosciences, dear Reviewers,

thank you again for the handling of the manuscript and for providing us your comments. We did our best to address them.

Please, find our answers in the table on the next page. We also uploaded a new version of the manuscript with tracked changes and another one with all tracked changes removed for further processing.

With kind regards and on behalf of all co-authors,

Marlene Dordoni (PhD student)

| Reviewer 2 | |
|---|---|
| The revised manuscript has been improved a lot by the author, but I still feel that the results and discussion part could be more straightforward and concise. Besides, I have some minor suggestions as below. Once these changes have been made, I fully support publication of the manuscript in Biogeosciences. | We thank the reviewer for his suggestions.

We now provided a better structure of the Result and Discussion section, with smaller paragraphs and references to figures (as substructured in for instance figure 2(a), (b) and (c) ). |
| L24-25: Put "(0.01 to 1.3 μmol L-1 d-1)" after "calculated turnover rates". | Done. |
| L42: Please make a definition of OM here, organic matter?
L46: Also, OM here. | Done. |
| L145: Again, the isotope mass model could be explained in more details, e.g., how the equation (2) was deduced? | Equation (2) in the main text is a re-arrangement of the following mass balance:

$$\delta^{13}C_s = \frac{(n_t \times \delta^{13}C_t + n_{fromOM} \times \delta^{13}C_{OM})}{(n_t + n_{fromOM})}$$

Here the subscript $\delta^{13}C_s$ refers to isotope compositions at any given sampling after time 0; the subscript "t" refers to time 0 concentration and isotope values; the subscript "OM" refers to organic matter sources (auto-POC, DOC, SED or allo-POC).

This information was added to the main text and further details regarding equation (2) are provided in the supplementary material S1. |
| L188: Is it DIC concentration? | Yes; we now made it clear in the text. |
| L295: The conclusion of significance of this study is weak and could be improved. | We think that the conclusions offer to valid and new points
    a) That the isotope method is able to produce plausible turnover rates;
    b) That our study outlines the fragility of the system especially in case of higher carbon loads.
Both of these points have been formulated more clearly. In addition, future work based on our results are recommended in the Conclusions. |
| | |